# Spatial Interactions in Business and Housing Location Models

Katarzyna Kopczewska *, Mateusz Kopyt and Piotr Ćwiakowski

Faculty of Economic Sciences, University of Warsaw, 00-927 Warszawa, Poland; mkopyt@wne.uw.edu.pl (M.K.); pcwiakowski@wne.uw.edu.pl (P.Ć.)
* Correspondence: kkopczewska@wne.uw.edu.pl

**Abstract:** The paper combines theoretical models of housing and business locations and shows that they have the same determinants. It evidences that classical, behavioural, new economic geography, evolutionary and co-evolutionary frameworks apply simultaneously, and one should consider them jointly when explaining urban structure. We use quantitative tools in a theory-guided factors induction approach to show the complexity of location models. The paper discusses and measures spatial phenomena as distance-decaying gradients, spatial discontinuities, densities, spillovers, spatial interactions, agglomerations, and as multimodal processes. We illustrate the theoretical discussion with an empirical case of interacting point-patterns for business, housing, and population. The analysis reveals strong links between housing valuation and business location and profitability, accompanied by the related spatial phenomena. It also shows that assumptions concerning unimodal spatial urban structure, the existence of rational maximisers, distance-decaying externalities, and a single pattern of behaviour, do not hold. Instead, the reality entails consideration of multimodality, a mixture of maximisers and satisfiers, incomplete information, appearance of spatial interactions, feed-back loops, as well as the existence of persistence of behaviour, with slow and costly adjustments of location.

**Keywords:** business location; housing valuation; density; multimodality; urban structure





## 1. Introduction

Despite the immensely large range of theories and empirical evidence in regional and urban studies, there are still very few explanations of the links between business and residential location within cities, with particular insight into firms' profitability and housing valuation. Existing theories of the location of both business and housing, together with their inherent agglomeration and spatial externalities effects, have, in fact, the same drivers and deserve to be considered jointly. Location factors affect where households and businesses decide where to live and locate a firm, respectively. Independent of the adopted research perspective and the decision-makers' internal motivations, the final outcomes are the densities of business and housing and the spatial distribution of economic activity reflected in the prices of real estate and the profitability of businesses.

The intersection of both streams, business and housing location and valuation, lies in their similar spatial determinants, and the spatial consequences of both processes in terms of clustering, density, agglomeration externalities, central business district (CBD) location, rural vs. urban sites, accessibility, proximity to important neighbours, quality and usefulness of neighbourhoods, absolute and relative locations, local and global perspectives, spatial segregation, etc. Both processes are interconnected. Real estate prices are a cost to businesses that impact their profitability. Conversely, prestigious neighbourhoods of well-performing companies increase the price of real estate.

An overview of the literature shows that both business and housing location and valuation models follow the same mainstream theoretical influences, from classical, behavioural, new economic geography (NEG), to evolutionary and co-evolutionary. However, their

developmental trajectory has been different—in business location theories, scholars abandoned the old concepts when shifting to new ones, while in housing, they have added new components to a classical base. This asymmetry in development has caused current models to be poorly integrated because of fundamental differences in assumptions. Secondly, both theoretical streams do not fine-tune their assumptions on spatial patterns, for example with respect to spatial interactions between agents, a mixture of maximisers, satisfiers and random decision takers, discontinuities in distance-decaying gradients, spatial interactions in determinants of processes or polycentric urban structures.

The remainder of the paper is structured as follows: In Section 2 this paper compares business and housing location theories and provides a unique insight into the spatial context to highlight common elements. We propose the following hypotheses: first, that the theoretical spatial assumptions of location models are mostly not fulfilled, and second, that neither of the models is out-of-date. At the same time, their joint application may shed some new light on occurring interactions. We suggest that regional theory requires an integrating umbrella across the agents and mechanisms involved. We show that localisation models in classical, behavioural, new economic geography, evolutionary and co-evolutionary frameworks, should be considered jointly, as they each account for different phenomena included in the von Thünen, Loesch, Weber, Alchian, Alonso, Tiebout, Webber, Krugman, Boschma and Gong and Hassink models. In Section 3, we discuss spatial issues of location theories including location changes, agglomeration effects, economies of density, bid-rent curves, spatial interactions, spatial externalities, a rationality-based mixture of satisfiers and optimisers, multimodality, and feed-back loops. Section 4 presents quantitative methods to capture these effects. In the empirical component (Section 5), we use spatial quantitative methods to verify existing relationships between business location and profitability and housing valuation. We hypothesise that there are visible links and spatial interactions between both spatial patterns. In an empirical illustration, we use geo-located point data for firms and real estate in Warsaw, Poland. We close the paper with conclusions and discussion (Section 6). An appendix provides empirical and technical details of the study.

## 2. Location of Business and Real Estate—Determinants and Decision Processes

### 2.1. Theories on Firms' Location

Over the last 200 years, beginning with the first business location model by von Thünen [1], many concepts have been proposed and many studies undertaken on how firms choose their location. The oldest, classical, approach was based on entirely rational optimisation, i.e., when selecting a place, firms seek to maximise their profit or minimise their costs. The models of districts by von Thünen [1] indicate that the location chosen depends on the activity type and proximity to the city centre. Later models, such as those of Alfred Weber [2], Loesch [3] and Moses [4], imply that choice is determined by maximisation of profit or sales with regard to raw-material markets and transportation costs. The Hotelling model [5], and its development from line to circle location by Salop [6], considers the relationship between the location and a companies' pricing policy. However, in these models, companies do not take advantage of differences in product characteristics—they compete and assess the products only in the geographic dimension and treat products as perfect substitutes.

Though classical models have been justified from a micro-economic perspective and rationality paradigm, they have been widely criticised. Webber [7], in his critique in the late 1960s, listed several categories of problem. The "error criticism" emphasizes that non-economic factors, such as prestige, persistent social networks, costly and imperfect information about alternatives, personal motivations, etc., may play a significant role in choosing and changing location. Simonian bounded rationality, driven by the impossibility of obtaining complete information and uncertainty about the future, causes agents to strive for a satisfactory level of profit or utility but not for a maximal one. The "deterministic criticism" implies that stochastic models will perform better than deterministic ones, mainly because of the inclusion of risk, unknown variables, and future uncertainty. Interestingly,

some current models still assume a maximisation mechanism regarding location, which suggests that there are doubts over the criticisms levelled above [8].

The criticism by Webber [7] presaged a wave of new models. The classical models were replaced by behavioural models of location and production, initiated by Pred [9], and concentrated on location dynamics [10]. Pred [9] considered two dimensions: (i) the availability of locational information, and (ii) the ability to use this information, which prevents firms and any locational decision-takers from operating as *Homo economicus* and being fully rational. Behavioural models assume that firms are bounded rationality satisfiers who decide to locate in a spatial margin of profitability, which is the area where incomes exceed costs, which can usually be determined after choosing the location (ex-post). Thus, the models concentrate on achieving goals rather than maximising profits. This change occurred to both assumptions and core principles, with interest in location changing to an interest in behaviour. Those models were applied until the mid-1980s [11]. They were replaced in the late-1970s by the emergence of the structural Dixit–Stiglitz–Krugman model, which emphasized market accessibility, and later by new economic geography (NEG) [12], anchored in agglomeration, economies of scale and transportation cost, explaining spatial concentration and, in consequence, persistent regional economic disparities. However, the evolution of these models changed the focus of interest; NEG is a trade-oriented, not a location-oriented approach. In NEG models, the selection of location impacts the business's profitability, while places and firms are heterogeneous [13].

Currently, NEG is being displaced by evolutionary economic geography (EEG) [14], which emphasizes contextual reaction over maximisation processes, as well as the importance of organisational routines, regional diversification, path dependence, lock-ins, and related and unrelated variety, over economies of scale and agglomeration issues. Its roots are in Alchian's adoptive and adaptive environment of the 1950s, which included uncertainty. Its potential successor, co-evolution in contemporary economic geography (CEG) [15] stresses the non-economic factors (e.g., institutions, rules), co-existence of industrial and non-industrial structures, and contextualisation. Both the EEG and co-evolution in CEG perspectives examine the place of activity not from a locational perspective but rather from a widely understood environmental and neighbourhood one. When applying EEG and CEG, which underline the significance of co-location patterns of business and place emphasis on tracking its dynamics, one should be aware of mock adjustments, which can be observed. Especially in urban geography, hundreds of years of history of cities have caused the delineation of market squares, the most important streets, quarters and districts, the location of authorities, arts and culture institutions, historical monuments, as well as the private ownership of flats, houses, land, etc. Thus, with contemporary data, one observes only local frictions in an urban location, while the general pattern, established in the past, is very persistent. However, agents in EEG also have bounded rationality, not because of a lack of information, but rather as a result of being focused on a comfortable path of development and the avoidance of risk.

The latest location studies exploit all the concepts which have evolved over the last 80 years. Fitjar & Gjelsvik [16] present an evolutionary and co-evolutionary design perspective and develop a model of local business-academia cooperation ("localised knowledge spillover"), which provides an explanation for local networks. For knowledge transfer, distance generates costs and the risk of losing the information during transference. The involvement in local clusters, long-term local cooperation, and contribution to the local community, even though it appears similar to EEG, can be found in Webber's critique [7]. The same is true about goals, as the firms in Fitjar's study are satisfiers as in the model developed by Claus & Claus [10], who do not optimize but achieve goals which might be suboptimal ("If the local university can make a useful contribution, firms might choose to look no further").

*2.2. Theories on Real Estate Location*

Similar to business location theory, which commenced with the profit-optimisation approach, the housing location theory introduced by Alonso [17] was based on differentiated bid-rent curves for specific land-use types (e.g., housing, commercial, and industry) within the city. Alonso claimed that different actors, such as households, industries, commercial establishments, etc., compete for locations within the city. They maximise utility from quick accessibility to the city centre with respect to available budget constraints. He assumed that centrally located land is more expensive than peripheral land, businesses can pay more for land than households, that rich people need less quick access to the city centre, and that high utility from accessibility is the concern of the poor. His urban-land-use segregation model is fundamental for job-housing balancing [18], assuming that people will be willing to live near to work if they can afford it. As summarised by Straszhem [19], residential location decisions are correlated with housing prices. In general, they are the function of urban spatial structure, including city size, population density, rent gradient, and housing location and business, as well as the budget constraints of households. Demand for real estate fluctuates because of the gentrification process and the rent gap, which appears also to be a driver [20,21].

Housing models have evolved somewhat differently than business location models. In contrast to business location models, they have never rejected the classical principles of optimisation formulated by Alonso. The new generations of theoretical models of location have only added new elements to the fundaments of the classical theory.

Behavioural housing models, as developed by Wheaton [22], imply that the Alonso model may not work if factors other than accessibility, such as income, taxation, the social composition of the neighbourhood, etc., impact the utility of housing location and drive settlement decisions. They have been extended into non-spatial models (such as the DiPasquale–Wheaton model) and explain housing demand with a depreciation rate of stock or rental price [23,24]. However, current behavioural models [25] optimise the residential location choice combined with transportation networks and still assume that endogenous neighbourhood characteristics determine it. In the behavioural approach [26], location decision is optimised with respect to housing prices and personal income; the neighbourhoods are assumed heterogeneous, while the primary decision driver is aspiration level. Inequality of income is the driver of spatial segregation of inhabitants.

NEG models [27] deal with competition between housing and business, which is strongly affected by pollution effects and includes agglomeration effects, which consequently establish industrial and residential clusters. Within the group of NEG models, Partridge et al. [28] tested urban cost-gradients, typical for distance-decay optimisation models, but under assumptions of hierarchical urban structure and different sources of agglomeration spillovers. Glaeser [29] built a complex NEG model of urban spatial equilibrium with three actors—firms, inhabitants, and developers—adding to it issues of agglomeration, wages, and transportation, and merging it with the Alonso–Muth–Mills model. Suedekum [30] extended Krugman's NEG model by incorporating the home goods sector and living costs.

Co-evolutionary models, such as [31], consider the relationship of the Alonso model to neighbourhood quality, primarily the Tiebout-model-driven quality of schools. Even though formally the EEG and CEG approaches appeared in the early 2000s, the old definition by Papageorgiou [32] that the "city is a complex system of spatial interdependence between its constituent elements: households, businesses, industries, and public institutions" created the opening for them, treating the other drivers of location as spatial externalities [33]. Taking a life-cycle perspective, household activity demand and accessibility to activities, such as employment, education, shopping, or medical services, may impact housing prices in urban spaces. However, an example provided by Beijing [34] showed that employment accessibility was the primary determinant of price and confirmed that housing and business locations were strongly connected. There is, further, a vast literature on location and neighbourhood quality in housing consumption from the perspective of

economic and socio-cultural changes in urban and metropolitan housing market areas [35]. Current studies [36] have confirmed the positive value of cultural amenities for housing markets when studied jointly with the effects produced by green areas, public transport, and university proximity. The complexity of the problem shifts the interest from urban landscape elements to well-being in cities as well. Though studies have confirmed the spatial nature of neighbourhood well-being and significance [37], they are inconclusive. They cannot explain why living in the city centre does not produce full happiness [38].

## 3. Spatial Issues in the Revision of Business and Housing Location Theories

The selection of business location in any location theory is inseparably linked to the location changes. In consequence, the classical optimisation approach implies that firms should move, preferably from worse to better locations. This can be found in Tiebout's [39] "voting with feet" concept for firms. His work on business location is not far conceptually from the much better-known idea of optimisation for taxpayers who benefit from public services [40]. Tiebout [39] links the place of business with its profitability and argues in favour of classical maximisation-based models. Even in cases of the random selection of locations by industry, the market competition mechanism will eliminate inferior firms from the market, which means that only well-located firms will survive.

Consequently, this causes the emergence of the profit-maximising agglomerative spatial pattern of location. However, this easy portability of business is in contradiction to the "sticky location" approach proposed by Webber [7], who claims that a business will prefer to stay in its location because of social and business links, knowledge of the market, suppliers, and customers, etc. Later models and studies have treated the stickiness of location differently, most often including it together with other assumptions and rarely considering the mechanisms.

Amongst well-established drivers of location changes are agglomeration forces. In fact, all location models, from von Thünen circles to the co-evolution economic geography (EG) approach, refer explicitly or implicitly to agglomeration effects, which are operationalised further by distance and density. Different mechanisms have been proposed for agglomeration: in the von Thünen model, it results from urban economies; in NEG it stems from transport costs and firm-level scale economies [41]; while in EEG it is the "endogenous co-evolution of agglomeration of industries and local formal and informal institutions, that are specific to certain industries and places" [15], and its existence is simply assumed. The well-established division into Marshallian specialisation and Jacobian diversification agglomeration externalities has been tested and discussed [42], and case-based evidence has been produced [43,44].

The latest studies shift the key issue of agglomeration phenomenon from production structure to economies of density, with greater emphasis on the spatial dimension in the urban context. As in [8], the seller maximises the profit through the "exploitation of the economies of density". The importance of density and urbanisation for agglomeration externalities find confirmation in study of [45], which showed that, "the threshold of urbanisation at which diversity, density and competition agglomeration externalities all generate positive effects was 33%". With more of an EEG flavour, the study by de Matteis et al. [46] found that for exporting firms, independently from internal features of the business, the most crucial factor for export was the local environment, including distance from foreign markets, the level of social capital, the efficiency of the public sector, the degree of financial development and the extent of agglomeration economies. The same regional agglomeration economies were confirmed by [47], which showed that these economies do not depend on the firm's characteristics. The first consequence of agglomeration are clusters, which have been well-studied since Porter [48] and developed by McCann [49]. The latest studies confirm that the optimal location for a business results from being part of a local cluster. This adherence may increase survival chances in times of crisis [50].

The above-presented overview shows that each of the mentioned streams, e.g., optimisation-based classical models, behavioural models, agglomeration-driven NEG, and

neighbourhood-driven EEG and CEG, are partly right but only shed light on some aspects of this complex phenomenon. Integration of these models could explain the spatial pattern of location more comprehensively.

The same problems occurring with business location models have been examined in housing models. Urban agglomeration effects are primarily modelled for firms, and housing appears as an extra component explaining the costs of living [30] or wage premium, or by adding an "Alonso" component for land consumption [41], all contained within the NEG framework. Agglomeration externalities traditionally arise from the central business district (CBD) issue, while new studies in the USA suggest that sub-centres may also become their source [51]. This can be optimised by defining new cores in a polycentric urban structure [52].

Change of location is mostly modelled with residential location choice, which is operationalised by binary-choice or multi-choice econometric models with respect to land-use and transportation, accessibility and commuting issues [53,54]. This approach was initiated by McFaden [55] and assumes a rational consumer, who "will choose a residential location by weighing the attributes of each available alternative and by selecting the alternative that maximises utility". In fact, it is based on gradients and Alonso's vision of the utility of households. The classical issue of bid-rent curves has been analysed by testing urban accessibility. The intra-city accessibility and its relation with housing prices and household income may be modelled in many ways [56]. Gutiérrez-i-Puigarnau et al. [57] found that the richer the household, the closer the residential location to Denmark's workplace. There are also many studies on the role of transportation in real estate valuation and household preferences [58,59]. Wong and Ho [60] proposed a housing location choice model considering a continuous transportation system for a city of arbitrary shape.

In general, the spatial behaviour of agents (people or firms) can be modelled theoretically using spatial interactions and spatial externalities. Both concepts previously appeared in Tobler's theory [61], e.g., "Everything is related to everything else, but near things are more related than distant things". However, it does not distinguish whether objects behave interactively (spatial interactions) or whether some are passive and simply influenced by other agents (spatial externalities). The theory of spatial externalities by Papageorgiou [62,63] does not assume interactions between agents. However, such effects decay with distance. There exist two surfaces: the population surface, where each agent is exerting and being influenced by some effects, and the externality surface, which is the total effect of all effecting agents over space, depending on the agents' distribution. Spatial externalities are, in fact, a kind of multiplier of benefits and costs for distance effects in the process of utility maximisation. The reaction of agents to change in location may differ, depending on stationarity or movability of the impact emitting point, but with respect to their own situation only and without insight into others' utility. A different approach, where absorption of externalities changes an agent's behaviour, was introduced by Papageorgiou [64]. It assumes that social interactions, including communication and constant comparison (re-assessment) of the spatial distribution of welfare opportunities, induce agents to move, and more specifically, to agglomerate. Both approaches to spatial interactions and externalities assume that agents, independent of interactions, are maximisers. Furthermore, spillovers are distance-decaying, while the urban spatial structure is unimodal. These are strong assumptions, which, when relaxed, may give different results.

The above-mentioned spatial issues generate three major problems. The first problem of spatial externalities and spatial interactions is the case of satisfiers, agents with Simonian bounded rationality and targeted to achieve a given level of wealth (as in Webber's [7] critique), instead of continuous maximisation as in Papageorgiou's models. They are affected by discontinuity over space and with a permanent steady state because some agents simply stop reacting after having achieved an internal goal. Agglomeration in neo-classical models is the effect of permanent spatial adjustments and re-location patterns and appears as the agents continuously look for more. When agents lose their motivation for permanent improvement, they stabilise. This was confirmed in knowledge-diffusion studies [65]. A variety of possible

behaviours can be observed in the whole population. Thus, theoretical and empirical studies should assume some proportion of utility optimisers and include satisfiers as well as random and undecided agents. This will cause a discontinuity in the spatial distribution of preference. Simultaneously, the break in a distance–decay function may be the reason for random spatial allocation, which depends on past positions, as in EEG.

The second problem in the explicit modelling of distance-decaying spatial effects is the multimodality of urban spatial structures. Even though the literature already addresses the issue of the existence of more than one core within the city (since Fujita & Ogawa [66], later in [67]), most models still explicitly or implicitly assume that there is only one urban core, in which the externalities start and spillover smoothly over space in a linear or negatively exponential way. The current models applied to multimodal urban structures are not as conclusive as for unimodal structures. Multimodality, and potentially reducing the unimodal spatial urban structure, is a safer starting point than the opposite strategy. In consequence, for multi-cores modelling, the distance–decay function might be stepped, not strictly decreasing. However, all patterns may be biased with spatial discontinuity, which is understood here as a limitation of the pure geographic patterns that deregulate gradients or distance–decay. It may be caused by administrative borders, by the non-linearity of processes, as well as by the non-optimising behaviour of agents (i.e., Simonian bounded rationality) and the overlap of the effects of gradients from different spatial processes.

The third problem is the iterative reaction of agents on the observed situation, i.e., a feed-back loop. Interactions by Papageogiou [64] are simply the possibility of comparing agents with others. Here there is no feed-back effect which would change the level of external effects emitted because of a change in some other agent's position. Papageorgiou's [64] model concerns direct impacts only, where feedback does not transfer between agents. However, many studies have observed, both for firms and real estate, the existence of an indirect effect, which allows for the transfer of feedback among agents. Although for a real estate market, the modelling of indirect effects is a standard [68], this is not represented in business location models. The existence of a feedback loop implies an inter-dependence in space. Spatial interactions can be observed at the level of persons and units, while they can also appear between markets as joint distributions. Many spatial econometric studies using spatial Durbin models [69] have convincingly shown that spatial inter-dependence of phenomena is often observed, and that direct and indirect spillover effects should be included in modelling.

Spatial analysis requires an integrated approach, both for individuals and aggregates, at a micro and macro scale, as a result of the diversity of agents [70]. Somewhat naïvely, but convenient for quantitative studies, the assumption is made that discovering the empirical shape of rent gradients and patterns of spatial accessibility with different modes of transport will allow for tracking of the trade-off between transport costs and housing price [70]. This, of course, follows from assumptions on unaffected, regular spatial distance-decay or cost-decay patterns, both for place attractiveness and human preferences. However, an overview of city housing and business location theories suggests that this simplification may go too far. Both spatial discontinuities and the spatial inter-dependence and agglomeration forces prevent us from assuming that spatial distributions are unimodal. Consequently, it is necessary to consider the multimodality of spatial patterns, together with the multi-surface problem, as in Papageorgiou's surfaces. Discovering composite spatial patterns, varying between subgroups, entailing the composition of different spatial layers with extreme values and discontinuities, is a complex task and should be based on a mixture of neoclassical, behavioural, institutional, and evolutionary approaches. Additionally, the issue of MAUPs (modifiable areal unit problems) should be considered, along with data aggregation and global and local perspectives, as well as the conditionality (vs. the unconditionality) of processes.

## 4. Spatial Methods to Verify the Spatial Mechanisms

### 4.1. Spatial Variables Included in the Modelling

Our research approach is that of "theory-guided factors induction" being midway between purely inductive and deductive modelling [71]. The explanatory variables are theory-connected, selected critically as potential candidates, however in an unstructured way and without specification of possible interactions. In this approach, less significant variables cannot simply be dropped, as they are anticipated to contribute to theory and should be considered [71].

We discuss the general types of spatial variables included in modelling. Each of them has its roots in theories discussed in Section 2 or Section 3. The kinds of spatial variables mentioned below apply to three separate datasets of geo-located point-data: for business, housing, and population. Typical pure point data are supplemented with additional spatial information, including values in a neighbourhood, distance to the core, and density around.

Value in the neighbourhood of a point is the average of values calculated within a given radius of the analysed point (firm location or real estate location) or the existence of firms from a given sector. We have calculated nearby profitability (*ROA_around*), nearby housing prices (*price_sqm_around*), and, as a dummy variable, the existence of firms from a given sector in close surrounding (*sec_XXX* for 28 sectors). They differ among the models in values and interpretations. *ROA_around*, in the model for *ROA_individual*, represents the average value of ROA in the surroundings of a given firm, and it becomes a spatial lag variable to measure spatial spillover. However, *ROA_around* in the model for *price_sqm* becomes an ROA average in the surrounding of real estate and tests the inter-dependence of local spatial distributions. The same applies to *price_sqm_around*.

Distance from a point to core location (as CBD) reflects gradients and the spatial distance-decay of phenomena. In the case of multi-core spatial organisation, the models should include distances to all cores. As in spatial interaction models, distance can be linear, multinomial, or logarithmic. The inclusion of multinomial distance allows for non-linearity to be dealt with flexibly. In this study, we measured the distances to two identified cores (*clust1* and *clust2*, derived with DBSCAN, described further) and applied the first, second and third power (^1, ^2, ^3) of those distances, which produced six variables (*dist_clust1^1*, *dist_clust1^2*, *dist_clus1^3*, *dist_clust2^1*, *dist_clust2^2*, *dist_clust2^3*).

The neighbourhood density can be expressed by a number of points within a radius of a point. Diverse local densities indicate agglomeration patterns and can detect cores of the city. Business density can be general (e.g., for all business units around), or sectoral (e.g., for a number of firms from a given or own sector). We included the following variables, which always represent a specific number of points within the analysed radius: *knn_business_density* (number of firms around), *knn_own_sector* (number of firms from the same sector around), *knn_BusinessServices* (number of firms from the business services sector around), knn_Wholesale (number of firms from the wholesale sector around). We have also included the local density of the population (*popul_around*) and the intensity of housing transactions (*no_of_trans_around*).

Beyond these individual data, the overall agglomeration within the territory can be tested with density functions based on points as well as on the entropy of the tessellated point-pattern (details in Appendix A, part 3). The areas of high density can be detected with the DBSCAN algorithm.

### 4.2. Data Used in Modelling

The empirical study was based on data for Warsaw, the capital city of Poland. We have integrated three datasets. The dataset on firms was collected from the AMADEUS/ORBIS database for 2016 and included approximately 24,000 geo-located point observations with information on ROA indicator (return on assets) and the business sector. A dataset for real housing transactions between 2012 and 2016 (a period of stable prices) was collected from the official register from the mayor's office and included approximately 36,000 geo-located point observations with information on price per sqm and general real estate

characteristics included in official documents (e.g., floor space, number of rooms, separate kitchen, balcony, cellar). The dataset for the population was released by Statistics Poland (https://geo.stat.gov.pl/inspire, 15 November 2021) as a 1 km × 1 km grid for 2011 with detailed data for number of inhabitants.

The original datasets were extended. For each observation of business data, we have calculated, within a radius of 500 m, the average profitability of firms, and the following density variables: the number of business units around, the number of firms from the same sector as an analysed firm, and the number of firms from the two most frequently appearing sectors, business services and wholesale. For each firm, we have also calculated, within that radius, the average transactional sqm price on the real estate market and the number of sales transactions, which creates the connection between these two datasets. We also transformed the gridded population into a point pattern by sampling within each 1 km × 1 km grid cell the points reflecting the population (1 point for 100 people). We have integrated this point pattern with business data by calculating the number of points (people) within a 500 m radius of each firm. The core locations of business were detected with the DBSCAN algorithm by searching a minimum of 1100 firms within a radius of ca. 150 m. Using this procedure, we have obtained two main density clusters (see Figure 1a). We calculated distances between each firm and the core of each cluster. All calculations were performed using R software (details in Appendix A, part 1). Descriptive statistics for the datasets are presented in Appendix B.

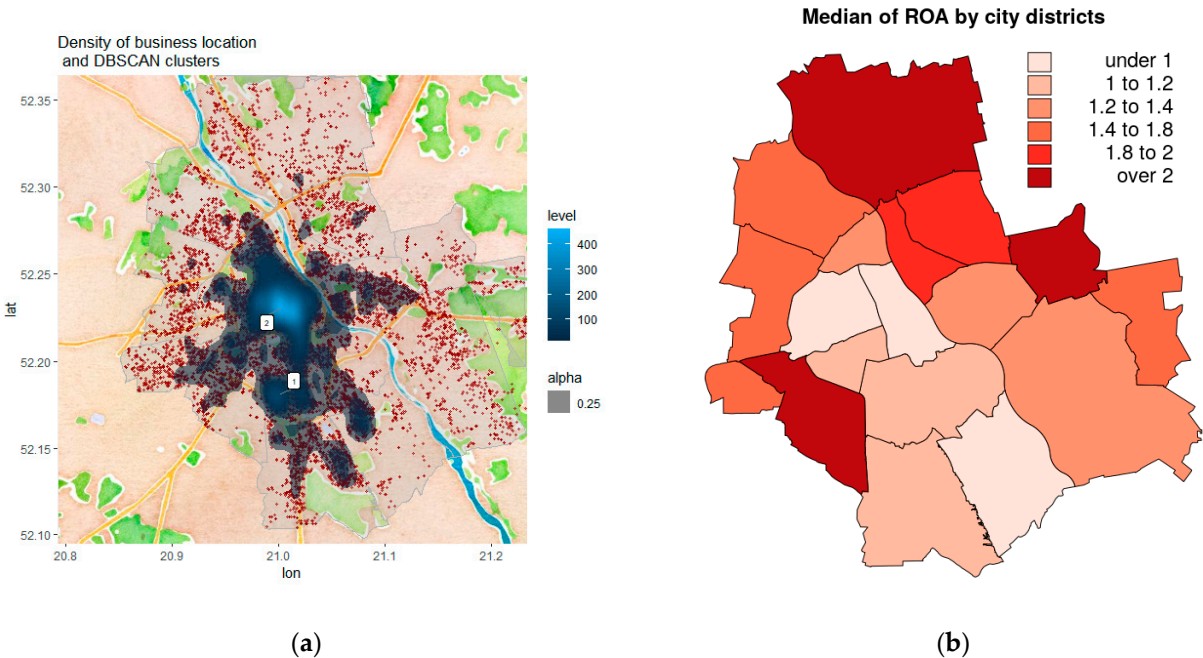

**Figure 1.** Business locations: (**a**) dense clusters of firms from DBSCAN to assess the multimodality of urban structure; (**b**) median values of ROA by city districts to assess the business profitability by locations.

Integration and comparison of point patterns require statistical solutions, which provide common references. For plotting, we used the rasterisation method, visible in Figure 3a,b. As conventionally required, the correlations of variables were also analysed to avoid multicollinearity in the model. In the case of geo-located data, this involved analysing the similarity of values of variables in similar locations. Typical correlation measurement was not effective in this case as it did not deal with space, while we operated with three geo-located datasets which included different observations. Therefore, we ran a more sophisticated algorithm based on the Rand Index (Figure 2b). Raw data (each variable individually) were clustered with k-means into 10 clusters, minimising within-cluster distances, and maximising between-cluster separation and group similar values.

For geo-located points, median cluster-IDs were calculated in raster cells, which showed the spatial patterns of variables. The presented raster has $50 \times 50 = 2500$ cells; there were 1926 empty cells, as no observations were recorded for these, and points appeared in ca. 22% (574) of cells. Within Warsaw's borders, there were 1625 cells, with 875 cells lying outside the city contour but within the bounding box. Warsaw's area is 517 km$^2$, so one raster cell covered 0.318 km$^2$ (ca. 560 m $\times$ 560 m). Thus, we obtained for each variable a raster with the median cluster ID assigned. Finally, we applied the Rand Index for each pair of variables, which analyses all possible pairs of pairs of observations (raster cells) and checks the percentage of pairs in the same clusters (Figure 2b), providing evidence of similarity. Rand Index = 1 means that all pairs of observations were in the same clusters in both partitions (for both variables). This method is suitable for data with different locations, as it smooths the local variability and includes a spatial location in calculations. Further details of the Rand Index are provided in Appendix A, part 2.

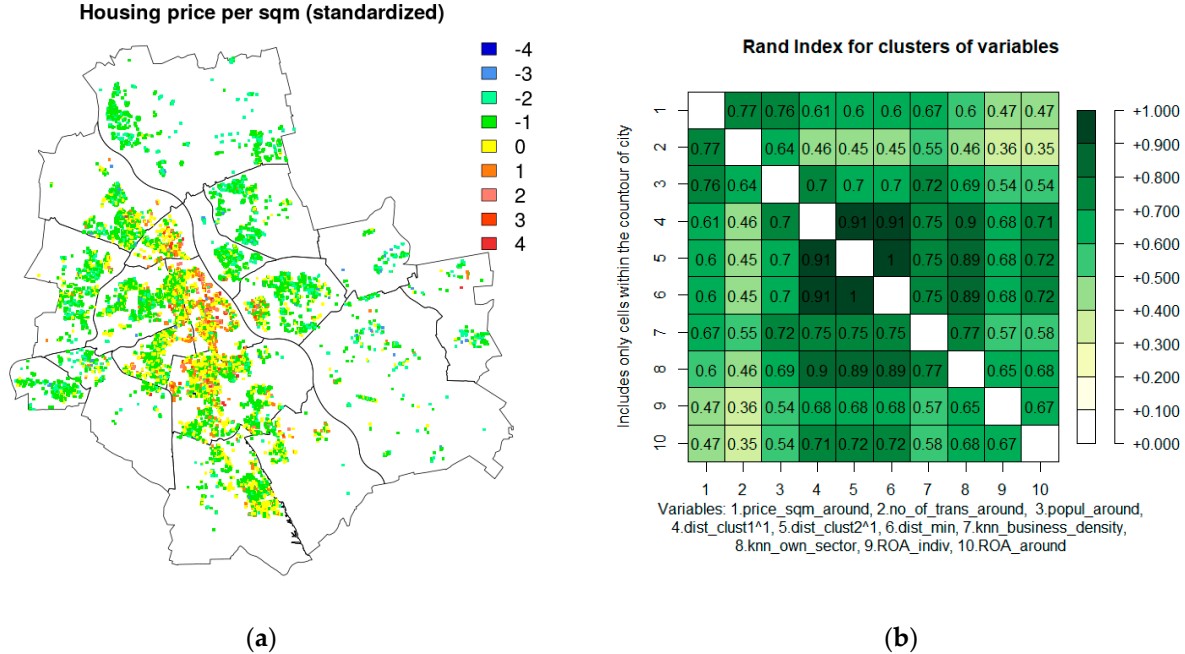

**(a)** **(b)**

**Figure 2.** Housing prices per sqm: (**a**) point plot of standardised data to assess spatial distribution of housing transactions; (**b**) Rand Index for k-means clusters of variables to assess the similarity of values over space.

In the housing analysis, we tested the impact of the available factors on price. The housing literature usually operates with two types of data: (a) offering prices, which usually come from web-scraped announcements, have a much broader description of features of real estate, but the price is not final; and (b) transaction prices, which usually come from city registers or legal contracts of purchase/sale, have a much narrower description of real estate, but the price is final. We operated with transaction prices, which automatically limited the possibility of explaining prices with internal features. We have checked the main drivers of 1 sqm price, and found that this mainly depended on the location (see Figure 2a), while other factors were of relatively minor importance.

### 4.3. Choice of Econometric Methods

The literature describes dozens of econometric approaches to model the housing market (Appendix A part 4). An overview prompts a number of remarks. Firstly, the method must answer the research questions; this is especially challenging when modelling implicit spatial patterns, as discussed in Section 3. Secondly, the method must be well adjusted to the data analysed; spatial effects can be captured with spatial variables or spatial estimation models using a spatial weights matrix, and doubling this information should be

avoided. Thirdly, explanatory econometric models (as in this paper) have different rules and problems than predictive models [72]. This relates especially to over-specification, goodness-of-fit, relation to theory, selection of variables, model selection and retention of insignificant covariates. The decision as to whether the model is intended to explain or predict impacts the econometric standards applied [72].

Below, we use econometric modelling to test mutual relations between housing valuation and business profitability in terms of the inter-dependence of the spatial distributions of these two processes. The modelling design allows for studying gradient issues (due to the polynomial distances), multimodality (due to the two cores), agglomeration patterns (due to the local densities), spillovers (due to the situation in the neighbourhood), and sectoral effects (due to the sectoral dummies). The central assumption is that of a feed-back loop and mutual relations between business, housing, and population. For that reason, we built several econometric models, for which the general form is as in Equation (1), while they differ in the dependent variable y, selected from the set of analysed variables:

$$Y = \beta_0 + \beta_i \cdot business.profits + \beta_j \cdot business.density + \beta_k \cdot housing.market +$$
$$+ \beta_l \cdot population + \beta_m \cdot distances.to.cores + \beta_n \cdot sectoral.dummies + \varepsilon \tag{1}$$

We estimated six models, with the following dependent variables: (1) transaction price of sqm of real estate (*price_sqm*); (2) profitability of a given firm (*ROA_indiv*); (3) average profitability of firms in the neighbourhood (*ROA_around*); (4) distance to first cluster (*dist_clust1*); (5) distance to second cluster (*dist_clust2*); and (6) local business density (*business_density*). All the models used a similar set of covariates, including business profits, *ROA_indiv* and *ROA_around*; business density (number of neighbouring firms within radius) *knn_business_density*; number of neighbouring firms from the same sector, *knn_own_sector*; number of neighbouring firms from business services and wholesale sectors, *knn_BusinessServices, knn_Wholesale*; housing market (average price of housing in neighbourhood), *price_sqm_around*; intensity of transactions in neighbourhood, *no_of_trans_around*; population (number of people living in neighbourhood), *popul_around*; distances to cores (polynomials of distance to core1 and core2), *dist_clust1^1, dist_clust1^2, dist_clus1^3, dist_clust2^1, dist_clust2^2, dist_clust2^3*); and sectoral dummies, to control if a firm from a given sector was in the nearby area (*sec_XXX*).

In the estimation procedure, we considered a-spatial (e.g., OLS, ordinary least squares) and spatial models. To assure the comparability of the six models and avoid doubling the information from the neighbourhoods, we needed to eliminate spatial lag models. Specification of the model already included spatial components including spatial lags as variables for the surrounding area and as distance to clusters variables. Thus, we considered OLS, a spatial Durbin error model (SDEM), and a spatial error model (SEM). As the data were geo-located points, the spatial weights matrix could be k nearest neighbours (*knn*) or inverse distance. We opted for knn = 5, as this often provides the best fit and/or lowest bias [73]. In the estimation procedure for three competing models, spatial models, due to the spatial variables included, were over-specified; the spatial parameters (theta, lambda) were negative, which means that they balanced the already captured spatial effect. AIC was similar in the spatial and OLS models. The spatial models also had far fewer significant variables than OLS. The OLS model was free from spatial autocorrelation of residuals (tested with Moran's I). On this basis, we concluded that the OLS models performed the best, and we selected these models for interpretation. The $R^2$ value of the final models (Supplementary Materials) varied, with the highest values (ca. 0.97–0.99) for equations explaining distance to core and business density, lower values (ca. 0.26) for housing prices, and the lowest values (ca. 0.05–0.1) for business profitability. This reflected the strength of the spatial patterns readable from the data. The $R^2$ value does not condition a good model in two situations: when the model is explanatory, not predictive [72], and when the analysis is conducted on populations, not random samples. In this study, we treated the model as explanatory, and we analysed the population having all registered housing transactions; this makes even low $R^2$ values acceptable. The adjusted $R^2$ (Adj.R^2

$= 1 - [(1 - R^2)(n - 1)/(n - k - 1)])$ was very similar to $R^2$, so for the sample size *n*, the correction due to the number of variables *k* was not necessary. For the same reason, the inclusion of many variables, insignificant in some models, did not lower the quality of the models [72] since the purpose was to ensure comparability and allow the model to follow "theory-guided factors induction" [71]. Due to comparability issues, we did not include real estate features in the model for housing price. This finds its justification in housing studies, which confirm that relative and absolute location are primary drivers of price, while internal real-estate features only fine-tune location-based valuations [74].

## 5. Interpretation of Results

### 5.1. Interpretation of Spatial Phenomena

The results of the regressions (Supplementary Materials) and visualisations (Figures 1–4) confirmed the existence of several spatial patterns, which we discuss below.

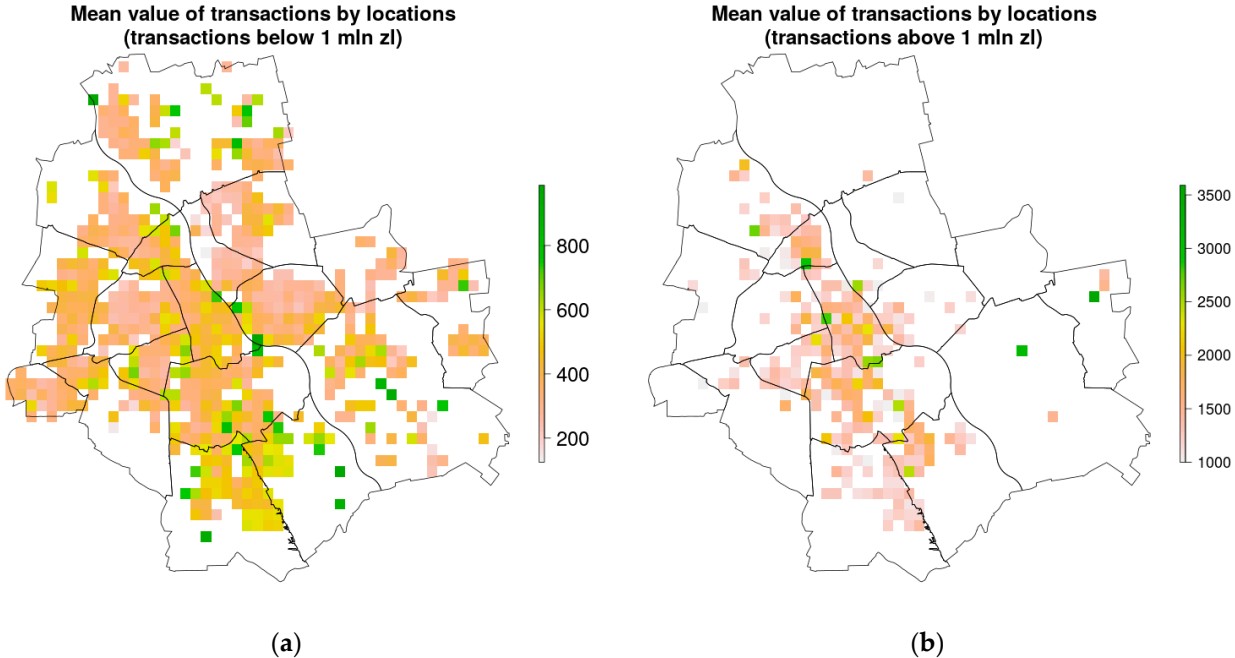

(**a**)           (**b**)

**Figure 3.** Spatial distribution of transactions by values: (**a**) below 1 mln PLN, (**b**) above 1 mln PLN to assess the spatial distribution of transactions and premium submarket location.

The existence of mutual relations between housing valuation and business profitability was evident from mutually significant coefficients of regression, e.g., β for housing valuation in model_3 explaining *ROA_around*, and β for *ROA_around* in model_1, explaining the sqm price of real estate. They also confirmed the strong inter-dependence of spatial distributions (considered above, x and y are average values in the neighbourhood), where the relation was negative. This implies that higher business profitability (ROA) coincides with lower prices of flats. This is confirmed by Figure 1b, where higher ROA is relatively peripheral, and Figure 3a,b, where higher flat prices are in the center. This relationship was also visible in the models for distance to clusters (model_4 and model_5) and for business density (model_6), where profitability and housing valuation showed a mutually significant relationship.

There was also a mutual relationship between housing valuation and business location. A significant regression coefficient was observed in model_1 (for housing valuation) with density of business (positive β at *knn_business_density*) and for distances to cores (some significant β, positive and negative). In contrast, β for housing price in the neighbourhood (*price_sqm_around*) was significant in models explaining business density (model_6) and

distance to cluster1 (model_4) and cluster2 (model_5). This implies that the higher priced flats were in dense business locations, especially in cluster 2.

**Tesselated business locations in Warsaw**          **Tesselated population locations in Warsaw**

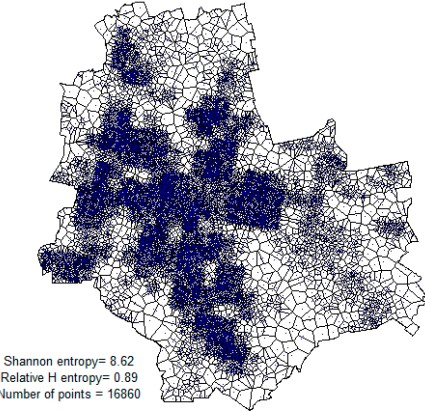

Shannon entropy= 9.38
Relative H entropy= 0.89
Number of points = 37197

Shannon entropy= 8.62
Relative H entropy= 0.89
Number of points = 16860

**Figure 4.** Tessellated business and population locations in the urban area of Warsaw to assess the degree of agglomeration.

Business location in the city reflects multimodality. Statistically, it was confirmed using a DBSCAN search for density clusters (Figure 1a). We found two significant and stable clusters: cluster1, located to the south, in a new business district, and cluster2, situated in the city center, next to the Old Town. Cluster2 was much broader than cluster1 and covered government buildings, major avenues, and so-called prestigious addresses. Distance to clusters was a significant factor in housing valuation and ROA in the neighbourhood; however, each cluster impacted differently, mostly because of the cluster characteristics. Both clusters had different business densities, which was evident from the DBSCAN statistics and diverse β coefficients for distances in model_6. Moreover, the clusters did not impact linearly, rather as the second and third power of distance in models.

The existence of clusters and diverse local densities result in the agglomeration phenomenon. In statistical analysis (Figure 4), the degrees of agglomeration of business and population were very similar and significant; the relative entropy relH of both point patterns equalled 0.89 (relH = 1 for uniform spatial distribution). The determinants of agglomeration were tested in model_6 for business density with many significant predictive variables. This phenomenon had spatial nature (significant β for distances), and was related to population, housing valuation and business profitability. However, while population density was associated with housing valuation (negatively), and business density (positively, business clusters were densely populated), it was not associated with business profitability. Agglomeration was also linked to sectoral effects: firms from the business services sector and wholesale sector preferred to locate more distantly from dense areas. A wider study of agglomeration, measured with entropy for tessellated point-patterns, presented in Figure 4, is available in [75]. Details of this method are presented in Appendix A, part 3.

The importance of distance means that a gradients and distance-decay pattern exist. Firstly, distance is significant for clusters in the business profitability model (model_3) and housing valuation model (model_1). As the β coefficients differ between clusters, there are no symmetric reactions over space, and distance-decay patterns vary in strength. The insignificance of some distance coefficients may also be considered as spatial discontinuity. Secondly, significant variables in model_4 and model_5 may be understood as gradients for clusters 1 and 2. The spatial patterns differ between those two clusters. For cluster 2 (centrally located, "old" business cluster), the closer to this cluster, the higher the flat prices, business, and population density, but the lower the ROA in the surrounding sectoral (business services and wholesale) density and the number of flat sales transactions. How-

ever, for cluster 1 (southern location, new business cluster), the closer to this cluster the higher the business and population density, but also the higher the ROA in surrounding and sectoral (business services and wholesale) density, and the lower the prices of flats and the number of transactions. Cluster 1 attracted, and cluster 2 repelled, the most dense sectors; the significant sectoral dummies in models 4 and 5 show which sectors tended toward which cluster. Interestingly, most industries were distributed in a mixed fashion, not around a given cluster.

The sectoral effects were visibly broader. The neighbourhood of three sectors (i.e., transportation, freight and storage, retail and metal production) lowered the housing valuation significantly by 3–4%, i.e., by comparing significant β coefficients (ca. −359, −293, −296 PLN) with a mean (8707 PLN) and median (8622 PLN) housing sqm price. The profitability of sectors was mostly well-diversified, with some positive effects in some sectors (e.g., business services, chemicals, food and tobacco, public administration) of 2.7 to 3.1 p.p. (for example, a negative impact in the utilities sector of 2.9 p.p.). Cluster 1 attracted sectors such as industrial, electrical and electronic products, metal products, and miscellaneous and utilities manufacturers (significant sectoral dummies in model_4). In contrast, cluster 2 attracted firms from the chemical and petroleum sectors, as well as waste management.

Finally, spillover effects could be observed when point observations were connected to their neighbourhoods. This was visible in business profitability where individual ROA was well-linked to ROA in the neighbourhood (model_2). This was also evident for the housing market, where higher prices were linked to a higher number of transactions.

### 5.2. Interpretation in the Light of Location Theories

The profitability of firms tended to cluster visibly. The spatial distribution was non-random, relative location mattered, and ROA in neighbourhoods was spatially correlated (β for ROA.around in model_2). In classical theory, optimisation of location with respect to gradients would be predicted to cause similar profitability of firms located next to each other. The spatial independence of profit indicates that classical gradient theory works well. In behavioural theory, information on the location is known, companies have different capabilities to relocate, and there should be a supply of the "best" places. Where neighbourhood dependence in profitability exists, (a) companies use the available information, or (b) there is the flexibility of mobility, or (c) there are over-profitable locations and the place of doing business is relevant. Firms and places are heterogeneous in the NEG approach, while a business's success depends on location and other internal factors. High profitability may be linked to location or agglomeration externalities. In the evolutionary approach, there is both a visible neighbourhood outlook and a contextual reaction to ROA (further information is provided in Appendix A, part 6). This implies that all theories can be applied to explain this phenomenon.

In general, sectoral profitability did not differ, as the sectoral β coefficients in model_2 and model_3 were mostly insignificant. This supports the viewpoint of classical theory that all companies are homogeneous. There is no basis for creating spatial clusters and optimal business territories from a behavioural perspective if there are no industry patterns. In the NEG approach, the importance of transport costs and market access should diversify sector profitability. This did not happen, with the limited exceptions of business services, chemicals, food and tobacco, and public administration. Thus, in general, the NEG fails, or the spatial scale necessary for these factors to differentiate business was too narrow. The EEG's flow of experience and tacit knowledge, strengthened with industry predispositions, would be predicted to create better-performing sectors, which was observable in only a few sectors. It appears that the assumptions of classical theory best fit the scenario. This method can approximate behavioural persistence, with adjustments to location being slow and costly.

The aggregated data indicated a slight difference in ROA by city district (Figure 1b). Surprisingly, the median and mean of ROA in core locations and CBD were lower than

those at the fringe. This is contrary to classical models, as the economic rent of location and spatial segregation was not observable. With homogenous firms but heterogeneous sectoral profitability, central locations should attract business with the highest marginal productivity. Similarly, in behavioural models, the location should reflect profitability's spatial margin, where incomes exceed costs. In NEG, when assuming different profitability of sectors, spatial agglomeration would reveal this difference. However, neither Marshall–Arrow–Romer specialisation nor Jacobian externalities are invisible in places of industrial heterogeneity. In the evolutionary context, one should expect a soft impact (as prestige) of the CBD. This method could reveal the city's structure, e.g., polycentric with two or more cores, or unimodal, with higher ROA modes. Detection of CBD would indicate the most attractive absolute locations. In the classical approach, the distance-decaying ROA would also evidence spatial gradients.

The location of firms revealed agglomeration in CBD (Figure 1a) and had diverse density, being much lower in the peripheries. In the classical framework, low transport costs in the centre dominate over the costs of such a location, which causes the agglomeration of many companies in the CBD. Behavioural bounded rationality may force the use of decision-making heuristics and follow the majority to the city centre. NEG's agglomerative forces empowered by agglomeration economies, transportation costs, etc., cause companies to strive for a central location. Finally, from an evolutionary perspective, the centre's surroundings are much more attractive than in other districts, hence the companies' high spatial concentration. The attractiveness consists of the benefits of density, the flow of knowledge, and the prestige of various institutions' neighbourhoods. The method reveals location structure with possible multimodality and agglomeration, and spatial discontinuities when the distance-decaying intensity of location appears.

Firms from selected sectors were well agglomerated in clusters (β coefficients in model_4, model_5 and model_6), while other sectors were distributed somewhat randomly. The aggregated data confirmed the classical assumption of a pure proximity pattern to CBD, while it failed because of multimodality in selected sectors. From a behavioural point of view, this supports Webber's critique [7] that the spatial concentration of sectors may be explained with non-economic factors as persistent social networks, the assumption that same-industry firms cooperate, and non-central locations appear because of the relevance of a stochastic model, or as a result of Simonian bounded rationality. In NEG, multimodality can be explained with market accessibility and the optimisation of transportation costs. The co-location of some sectors in NEG would underline Jacobian externalities. An evolutionary perspective covers both phenomena. Old centres attract some businesses because of prestigious neighbourhoods and proximity to institutions. At the same time, new centres also emerge as places grouping selected sectors and offering alternative neighbourhood externalities other than in historical centres. Co-locations depict the related variety and contextual reaction.

The prices in housing transactions followed clear spatial patterns; there were significantly higher prices in the CBD and lower prices in the peripheries (Figure 2a). In subgroups of transactions below and above 1 mln PLN (about 0.23 mln EUR) (Figure 3a,b), it can be seen that in housing for the rich (above 1 mln PLN) locations from the CBD predominated, while in most districts, such transactions did not occur. From the data analysis, we determined that the number of transactions was the highest in the CBD and in the cheapest district in the north. Transaction prices were strongly correlated to the neighbourhood. The higher the transaction price, the stronger the differentiation among neighbours. It was evident that the old classical Alonso bid-rent model works. However, the CBD was definitely for wealthy social groups, which is the anti-Alonso view. It is not clear if prices may have resulted from accessibility to the workplace. The behavioural model can be confirmed, as prestige locations were more expensive, mainly in the city's central area. This ensures the existence of a spatial margin of profitability when utility, instead of profit, is considered. The NEG approach can also be confirmed, as cost gradients and distance-decaying prices were found to exist, and transportation costs mattered mostly

for the rich. In an evolutionary context, the neighbourhoods, local networks, and prestige were evidenced, and the surrounding areas generated clustering. Significantly higher prices in the three districts suggest a polycentric structure and the impact of absolute location.

### 5.3. Discussion of the Results

This paper presents one of the first trials to quantitatively operationalise the comparison of location theories and their spatial effects as inter-dependence of different spatial distributions, studying gradients, multimodality, agglomeration patterns, spillovers, and sectoral effects. We deal with issues that have not been addressed until now and are challenging. However, we believe that many important aspects of business and housing location models have been revealed.

We have not sought to solve all issues in business and housing location but instead to shed some light on spatial aspects. Typically, economic or land-use models are simplifications, as the enormous complexity of reality is almost impossible to grasp in a single study. Secondly, we designed our research as "theory-guided factors induction", which means our approach was more theory-based than data-mining-based. For that reason, we have omitted many issues which are often included in popular deductive models, e.g., commuting, transport infrastructure, etc.

We also wished to underline the heterogeneity of economic agents, and especially their different attitudes to wealth and utility. Many economic models assume, for simplification, neo-classical full rationality and maximisation of utility. However, non-classical models stress the existence of bounded rationality and the co-existence of optimisers (i.e., rational agents) and satisfiers (i.e., those who reach a satisfactory level and stop improving their situation). Spatial gradients work well in mono-centric cities with rational maximisers as inhabitants. In mixtures with satisfiers and/or next to the urban core, the non-linearities and discontinuities appear. There are no studies on these aspects, while our approach can detect the appearance of these issues, e.g., through polynomial distances, multi-cores, and the insignificance of some variables. We feel that it is an important part of regional science worth studying.

### 6. Conclusions

This study has shown that theoretical approaches to business and housing location, even if they are related in reality, should be treated separately in analyses. The approach to the analysis of problems started with a classical framework, then behavioural, new economic geography, evolutionary and co-evolutionary models were applied, and, due to their relative independence, business and housing theories evolved in different directions. In the business location context, it is apparent that abandoned old concepts have taken on newer ones. Conversely, in housing location, new theoretical waves have been successfully added to strong classical foundations. Housing location models seem not to be subject to Webber's critique, which has though led to very different approaches in contemporary analyses in relation to business localisation models. This may indicate that the consequences of this critique were not as important as business location researchers thought. There is also an imbalance in existing studies: firms' location (with respect to profitability) has received more theoretical consideration, while empirical studies are less represented; conversely in housing location and valuation, empirical studies have dominated (especially hedonic models and residential location choice approaches), while theoretical studies are limited.

Secondly, both research areas have inherited spatial components, for which assumptions are rarely fulfilled, and as a result, models do not work as predicted. Reality demonstrates that assumptions on unimodal spatial urban structure, the existence of rational maximisers, distance-decaying externalities, and a single pattern of behaviour, do not hold. Instead, situations involve multimodality, a mixture of maximisers and satisfiers, incomplete information, the appearance of spatial interactions, feedback loops, as well as the existence of persistence of behaviour with slow and costly adjustments of location.

Thirdly, our paper's empirical approach has evidenced that none of the theories were entirely wrong but that none of these concepts could be taken as comprehensive. There is a need for at least five theoretical approaches (classical, behavioural, NEG, evolutionary, and co-evolutionary) to explain the location mechanisms in a complementary way.

Fourthly, in our empirical example, we have shown that there exist links between housing valuation and business location and profitability. We have confirmed the existence of all the highlighted spatial phenomena including agglomeration, gradients, spatial discontinuities, the impact of density, and multimodality.

We conclude that regional science would profit from integrating both processes and families of models. Housing valuation and business location have similar spatial patterns, and their determinants and mechanisms can also be congruent. We argue that none of the single theoretical models can work because of their high complexity. An "umbrella" theory is required that integrates all available concepts. In addition, there is a strong need to rethink spatial components and fine-tune them to observed mechanisms to discover all dimensions of spatial patterns.

**Supplementary Materials:** The following are available online at https://www.mdpi.com/article/10.3390/land10121348/s1. Table S1: Estimation of models.

**Author Contributions:** Conceptualisation, K.K.; methodology, K.K.; software, K.K., P.Ć., M.K.; validation, K.K.; formal analysis, K.K.; investigation, K.K.; resources, K.K., P.Ć., M.K.; data curation, K.K., P.Ć., M.K.; writing—original draft preparation, K.K.; writing—review and editing, K.K., P.Ć., M.K.; visualisation, K.K.; supervision, K.K.; project administration, K.K.; funding acquisition, K.K. All authors have read and agreed to the published version of the manuscript.

**Funding:** This research was a part of a project titled "Spatial econometric models with fixed and changing structure of neighbourhood. Applications to real estate valuation and business location" funded by the National Science Center Poland (Krakow, Poland) [OPUS 12 call, grant number UMO-2016/23/B/HS4/02363].

**Institutional Review Board Statement:** Not applicable.

**Informed Consent Statement:** Not applicable.

**Data Availability Statement:** Due to formal restrictions data are not publicly available.

**Conflicts of Interest:** The authors declare no conflict of interest.

## Appendix A

1. Technical details of estimation in R

We used the following R packages: sp, spdep, rgeos, spatstat, raster, ggplot2, ggmap, plot.matrix, RColorBrewer, fossil. For efficient analysis of surroundings, we used spatial geometry–rings (rgeos::gBuffer()), which were overlaid on point pattern (sp::over()) to indicate with dummy if the point was covered with geometry. Dummies were used to derive statistics of the neighbourhood–number of points or their features. Figure 1a was generated with ggplot2:: and ggmap::, we used as a background map a Google watercolour map of Warsaw in stamen format (using ggmap() and getmap()), on which we added point-pattern of business locations (using geom_point()), the administrative contour of Warsaw city (using geom_polygon()) and density layer of business (using stat_density2d()). Figure 1b was generated with plot(), and counts the median value of points overlaid (with sp::over()) on spatial polygons of Warsaw districts. Figure 2a used plot() and findInterval() to visualise values of points. Figure 2b was plotted with plot.matrix::plot.matrix(), while the Rand Index was calculated with fossil::adj.rand.index(), colours were derived with RColorBrewer::brewer.pal() and k-means clusters were found with stats::kmeans(). For Figure 3 we rasterised values with raster::rasterize(). On Figure 4, we show point-pattern, which was converted with spatstat::ppp() and tessellated with spatstat::dirichlet(), while entropy was derived as algebraic computation of areas computed with spatstat::tile.areas(). Core locations were derived with dbscan() and hullplot() from dbscan::. Distances between points and cores were cal-

culated with sp::spDistsN1(). Sampling within grid cells was done with sp::SpatialPoints(). Regressions from Supplementary Materials were done with stats::lm().

2.　　Rand Index as a measure of dependence of spatial point-patterns

The Rand Index assesses if two features have the same distributions. Each feature (i.e., single variable or set of variables) is clustered (e.g., with k-means). One check per given observation, for which both characteristics were measured, was classified in the same cluster. As comparing ID of clusters would be inefficient (because of, e.g., relabelling), one compares pairs of pairs of observations. It checks which clusters the pair of observations were classified with respect to the first feature and compares this classification with regard to the second feature. The Rand Index is expressed as:

$$R = (a + b)/(a + b + c + d)$$

and checks if pairs are in the same or different clusters: a—in t0 the same, in t1 the same, b—in t0 different, in t1 different, c—in t0 the same, in t1 different, d—in t0 different, in t1 the same; thus, the counter is always the same (a) and always different (b). Clusters, and the denominators are all possible outcomes (a,b,c,d). Rand Index = 1 means that partitions always agree (c and d are NULL) and clusterings are the same, while Rand Index = 0 means that partitions migrate and do not agree for even a single pair.

An example of the Rand Index is presented in Figure A1. The measure is independent of naming the clusters; thus, renamed labels do not interfere with the comparison of clustering similarities (see a comparison of sets 1 and 2). The measure is sensitive to spatial patterns changes (see a comparison of sets 1 and 3). The left part of Figure 1 illustrates cells (A, B, C, D) which belong to clusters (1, 2) in three sets: set 1 with an initial pattern, set 2 with re-labelling of clusters and no change in pattern, set 3 with a shift of clusters and significant change of pattern. The right part of Figure 1 counts the pairs.

The middle two columns of the table on Figure A1 are to compare set 1 and set 2. In general, there are six possible pairs to analyse: A:B, A:C, A:D, B:C, B:D, C:D. Both cells of pair A:B were in cluster 1 in t0 and in cluster 2 in t1; thus, this pair was classified as scenario a) (t0 the same/t1 the same). The same applied to pair C:D. However, cells of the other four pairs were always in different clusters, e.g., in pair A:C in t0 they were in clusters 1 and 2, and in t1 in clusters 2 and 1, respectively. This makes in total: two scenarios (a), four scenarios (b), and no scenarios (c) and (d). Rand Index, expressed as (a + b)/(a + b+ c + d), is 6/6 = 100%. This example shows that relabelling of clusters does not impact the classification.

The last two columns of the table in Figure A1 compare set 1 and set 3. In general, there are also six possible pairs to analyse: A:B, A:C, A:D, B:C, B:D, C:D. Pair A:B, which previously was in the same colour, now experienced a change: A:B were in the same cluster (1) in set 1 and in different clusters (1 and 2) in set 3. That is why A:B was classified as scenario (c) (t0 the same/t1 different). In fact, in sets 1 and 3, there were no cells that stayed in the same cluster—scenario (a), there were two pairs (A:D, B:C) which were always in different clusters—scenario (b) and two pairs (A:C, B:D) which shifted from different clusters in set 1 to the same cluster in set 3—scenario (d). Thus, Rand Index is (a + b)/(a + b + c + d) = 2/6 = 33%. This example illustrates that the Rand Index is sensitive to a shift in spatial pattern.

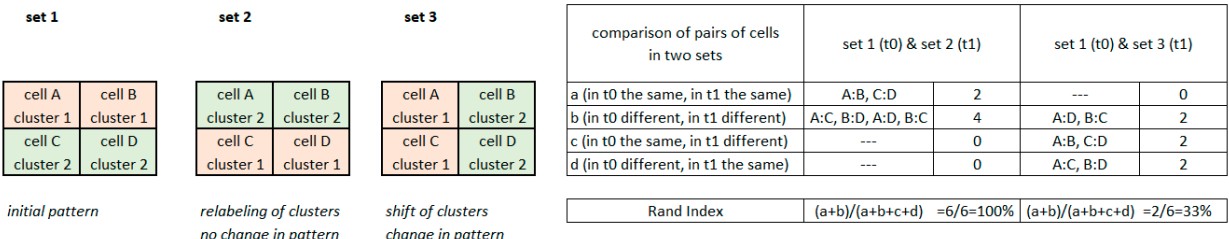

**Figure A1.** Example analysis of patterns with the Rand Index.

3. Agglomeration measure based on the entropy of tessellated point-pattern

Measuring agglomeration degree with the entropy of tessellated point-pattern was originated by Lews [76] and developed by Kopczewska [75]. The idea behind this is that in the case of no agglomeration, points are distributed equally on the surface, which makes the reference point for this measure. Any empirical point pattern can be tessellated with Voronoi's polygons, which divides the area continuously into non-overlapping tiles. Tessellation tiles are limited by lines located precisely halfway between two neighbouring points. Each tile has its area, and a total of all tiles' areas equals the area of the bounding region. Shares s of areas in the total area become a percentage measure, which is an input into entropy measure H.

$$H = -\sum_{n=1}^{N} s \cdot \log s$$

In the case of equally distributed points, all tiles' areas are equal, and entropy equals Hmax.

$$H_{max} = -\sum_{n=1}^{N} s \cdot \log s$$

A convenient measure to compare empirical and maximum entropy is relative entropy expressed as the ratio of both:

$$relH = \frac{H_{empir}}{H_{max}} = \frac{H_{empir}}{\ln n}$$

Thus, in the case of equally distributed points, without agglomeration pattern, relative H equals 1. Agglomeration is detected when relative $H < 1$.

4. Econometric methods in verification of location choice

Most existing studies use econometrics for the verification of theories on location choice. The popular hedonic model by Rosen [77] was in previous decades expanded and fine-tuned in terms of explanatory variables and estimation methods [78] (see Appendix A, part 5). This model allows for the valuation of characteristics of real estate and when spatial components are added, as a spatial weights matrix, spatial coordinates of a location, dummy for CDB, etc., measures the spillover effects of the studied phenomena and its determinants [79,80]. The key attribute of company or real estate location data is that they are geo-referenced, which means that their locations are related to the site and may depend on the features of the nearest areas [81]. The spatial distribution of companies and housing is by no means homogeneous, and a spatial approach helps differentiate the groups. In addition, the lack of consideration of spatial effects may contradict the theoretical assumptions, as in new economic geography, where agglomeration economies often appear mainly at the local level [82]. However, its popularity has meant that they are based on a plethora of determinants but do not produce the same conclusions. Meta-studies on hedonic models, such as [83], using OLS and quantile regression, seek to find the general rules on the sign and volume of marginal effect coefficients in models determining the real estate price.

The choice of residential location can be measured, following McFadden [55], with binominal or multinomial choice models for alternatives or their aggregates. The model weights the features of possible options and maximises the utility by choosing the best alternative. Kohlhase and Ju [84] tested, using a discrete-choice model, the firm-location assuming a polycentric city [85]. One can also apply multinomial mixed logit to model the housing choice and urban location parallel with respect to differences between individuals [86]. In the catalogue of non-spatial methods, the multiple criteria complex proportional assessment (COPRAS) can also be found based on weighted preferences. It has been applied to real estate's utility degree and market value determinants using experts' assessments of preference decisions on competing alternatives [87]. There is also a stream of multi-agent systems to model location optimisation's urban dynamics [88].

The last decade has brought the application of machine learning methods to location problems, e.g., the artificial neural network (ANN) [89], genetic algorithms [90], k-nearest neighbour smoothing [91], fuzzy logic [92], boosted trees, support vector regression and CART [93].

5.    Hedonic models for housing valuation

Parallel to residential location models, a stream of hedonic models was originated by Rosen [77]. They incorporate the different factors of real estate itself as well as of neighbourhood and spatial externalities on housing valuation. Originally, the hedonic model was based on the Alonso approach and designed as an operationalisation in search of market equilibrium with respect to budget constraints. Thousands of citations have demonstrated its enormous potential for empirical research on real estate. Current hedonic models [94] include economic factors (i.e., in relation to income and price or government tax policy), intrinsic factors (e.g., ownership, type of premises, size, age, condition, additional surface, floor, etc.), as well as socio-cultural issues (e.g., distance to relatives, job and public services, the reputation of the place, noise, crime, etc.). There are also models referring to government (e.g., urban, locational, housing) policy [95,96], ecological issues, or variations in prices and interest rates [97]. Geographical issues are included as the characteristics of local markets [98], existing land use patterns [99] or topography and proximity to networks [100], as well as the distance to the city centre or place of work.

6.    Links between the business location and its profitability

A separate issue in the literature on business is to link firm location with its profitability, measured by return on investment (ROI), on assets (ROA), on sales (ROS), and on equity (ROE). Profitability and location are often linked with other internal and external organisational factors, such as ownership, management, age, innovations, acquisitions, and associated management turnover [101], worker representation in unions and works councils [102], technological capability to profit on knowledge externalities [103], and capital structure [104]. SME surveys [105] have shown that firms located in cities are more likely to be profitable than those located in smaller towns or rural areas. Comprehensive empirical studies evidence that entrepreneurs consider many factors when establishing a new firm, which varies over space, e.g., economic policy, scope of product sales, industrial structure of countries and regions, distance to capital city, location of resources, demographic variables, localisation patterns, local demand and supply linkages, taxes and zoning system, sectors and segmentation, economic growth rates, per capita income, government policy, market size, institutional stability and the degree of economic freedom, infrastructure, institutional issues, the importance of market access and factor endowments, innovation capacity, clustering and agglomeration issues, consulting, territorial distribution of economic activity, etc. However, agglomeration externalities and elusive "good" locations do not guarantee high firm profitability [106]. Current applied studies on location show that explanation of business location within the framework of a single theory only is impossible. There is also the plethora of factors that "the probability of a plant start-up is more strongly related to increases in local market size and labour force qualification, lower labour costs, and a more diversified economic environment" [107], which, in fact, refers to all streams of theory. Long-term decisions depend on budget, costs, opportunity cost, and utility/profit.

Somewhat different is the integrated model of residential and employment location (IMREL), which considers where to locate new real estate from an optimisation point of view based on random utility theory with fixed travel costs. It was formulated and applied as a policy analysis tool in the Stockholm region [108,109]. IMREL uses a heuristic approach; the location choice model was solved sequentially with a user-optimal route choice model (i.e., trip assignment) to determine the location and travel choices of all workers in the Stockholm region. These choice patterns were then used to investigate the welfare maximising location of housing supply [110].

## Appendix B

**Table A1.** Descriptive statistics of the analysed dataset.

| VARIABLES | Min | Q1 | Median | Mean | Q3 | Max |
|---|---|---|---|---|---|---|
| lat | 52.104 | 52.196 | 52.227 | 52.221 | 52.239 | 52.363 |
| lon | 20.856 | 20.985 | 21.007 | 21.011 | 21.031 | 21.254 |
| ROA_indiv | −29.950 | −4.120 | 1.040 | 1.320 | 7.690 | 29.990 |
| ROA_around | −27.730 | 0.032 | 1.006 | 1.338 | 2.592 | 28.210 |
| knn_business_density | 1.000 | 23.000 | 60.000 | 130.473 | 201.000 | 888.000 |
| knn_BusinessServices | 0.000 | 4.000 | 13.000 | 29.948 | 45.000 | 234.000 |
| knn_Wholesale | 0.000 | 3.000 | 9.000 | 17.031 | 24.000 | 104.000 |
| knn_own_sector | 1.000 | 2.000 | 7.000 | 18.037 | 21.000 | 234.000 |
| popul_around | 0.000 | 8.000 | 19.000 | 18.212 | 26.000 | 58.000 |
| price_sqm_around | 2528.845 | 7709.940 | 8621.766 | 8707.044 | 9543.100 | 20,208.349 |
| no_of_trans_around | 0.000 | 3.000 | 30.000 | 53.285 | 87.000 | 526.000 |
| dist_clust1^1 | 0.147 | 4.714 | 5.986 | 6.503 | 8.015 | 20.607 |
| dist_clust2^1 | 0.086 | 1.461 | 4.073 | 4.598 | 6.771 | 18.291 |
| sec_Bank_Insur_FinServ | 0.000 | 0.000 | 0.000 | 0.046 | 0.000 | 1.000 |
| sec_Biotech_and_LifeScien | 0.000 | 0.000 | 0.000 | 0.006 | 0.000 | 1.000 |
| sec_BusinessServ | 0.000 | 0.000 | 0.000 | 0.212 | 0.000 | 1.000 |
| sec_Chemic_Petr_Rubber_Plast | 0.000 | 0.000 | 0.000 | 0.011 | 0.000 | 1.000 |
| sec_Communicat | 0.000 | 0.000 | 0.000 | 0.010 | 0.000 | 1.000 |
| sec_CompHard | 0.000 | 0.000 | 0.000 | 0.001 | 0.000 | 1.000 |
| sec_CompSoft | 0.000 | 0.000 | 0.000 | 0.042 | 0.000 | 1.000 |
| sec_Constr | 0.000 | 0.000 | 0.000 | 0.112 | 0.000 | 1.000 |
| sec_Food_Tobacco_Manuf | 0.000 | 0.000 | 0.000 | 0.010 | 0.000 | 1.000 |
| sec_Industr_Electric_Electro | 0.000 | 0.000 | 0.000 | 0.014 | 0.000 | 1.000 |
| sec_InformServ | 0.000 | 0.000 | 0.000 | 0.001 | 0.000 | 1.000 |
| sec_Leather_Stone_Clay_Glass | 0.000 | 0.000 | 0.000 | 0.003 | 0.000 | 1.000 |
| sec_Media_Broadcast | 0.000 | 0.000 | 0.000 | 0.026 | 0.000 | 1.000 |
| sec_Metals_MetalProd | 0.000 | 0.000 | 0.000 | 0.007 | 0.000 | 1.000 |
| sec_MiningExtr | 0.000 | 0.000 | 0.000 | 0.003 | 0.000 | 1.000 |
| sec_Misc_Manuf | 0.000 | 0.000 | 0.000 | 0.002 | 0.000 | 1.000 |
| sec_Print_Publish | 0.000 | 0.000 | 0.000 | 0.018 | 0.000 | 1.000 |
| sec_PropertyServ | 0.000 | 0.000 | 0.000 | 0.110 | 0.000 | 1.000 |
| sec_Publ_Adm_Edu_Health_SocServ | 0.000 | 0.000 | 0.000 | 0.046 | 0.000 | 1.000 |
| sec_Retail | 0.000 | 0.000 | 0.000 | 0.046 | 0.000 | 1.000 |
| sec_Textiles_Cloth | 0.000 | 0.000 | 0.000 | 0.003 | 0.000 | 1.000 |
| sec_Transp_Manuf | 0.000 | 0.000 | 0.000 | 0.002 | 0.000 | 1.000 |
| sec_Transp_Freight_Storage | 0.000 | 0.000 | 0.000 | 0.025 | 0.000 | 1.000 |
| sec_Travel_Pers_Leisure | 0.000 | 0.000 | 0.000 | 0.051 | 0.000 | 1.000 |
| sec_Utilities | 0.000 | 0.000 | 0.000 | 0.019 | 0.000 | 1.000 |
| sec_Waste_Manag_Treat | 0.000 | 0.000 | 0.000 | 0.005 | 0.000 | 1.000 |
| sec_Wholesale | 0.000 | 0.000 | 0.000 | 0.158 | 0.000 | 1.000 |
| sec_Wood_Furnit_Paper_Manuf | 0.000 | 0.000 | 0.000 | 0.006 | 0.000 | 1.000 |

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
