# Peer review of "Spatial Interactions in Business and Housing Location Models"

_land, doi:10.3390/land10121348_

Round 1

Reviewer 1 Report

Indicate to the authors that Table 1 does not appear either in the document or in the annexes.
It is indicated in the document that the results of the different OLS models are presented in Table 1 (p. 8) and where the table should be (p. 9) it is indicated to see the annexes. On the other hand, in Appendix A it is indicated that different R packages have been used as software to perform the calculations and the one used for the OLS is indicated, but the table with the results is not attached.
The percentages of the results are commented in the text, but they cannot be checked. As well as the coefficients of determination and significance of the variables.
Therefore, it is requested to attach it to complete the review. Thank you.

-------------------

25 October update:

  1. Table 1 (with the results) should be attached to the document on page 9 and not as an annex. In addition, a new table should be added describing the variables and indicating the descriptive statistics of the variables.
  2. The naming of the variables in figure 2b should be homogenised with the nomenclature used in table 1 with the results of the models. Furthermore, in model_3 the dependent variable has a different nomenclature from that indicated in the text and in the table itself.
  3. When table 1 with the results is available, the following issues can be observed:

-          In the models evaluated there is an overspecification of non-significant variables.

-          Models 1, 2 and 3, even with 43 independent variables, only explain 20%, 5% and 10% of the variability of the different dependent variables, respectively. (price, business profits, housing valuation, distances to clusters and business density).

Therefore, after analysing the results provided by the statistical analysis, conclusion number four would not be entirely correct as it cannot be stated that "in the empirical example, we show that strong links between housing valuation and business location and profitability", since the models in the case study explain a very small variability of the model and it is understood that the results are not generalisable.

Author Response

Dear Reviewer, sorry for technical problems, please see the missing Table 1 attached. 

best regards, Author

Reviewer 2 Report

It is considered to have good research results as a study  that integrates business and housing location models. However, There are too many variables used in the model, so I suspect overfitting. Further explanation is needed on these points.

Author Response

(The authors gave the same response as above.)

Reviewer 3 Report

Manuscript ID land-1435401 Spatial interactions in business and housing location models

A first impression of this paper is that it is a review of work in the fields of business location models and housing location models. The reference list contains 104 items. Indeed, much of the work reviews many contributions particularly in the field of business location models pointing out that housing location models are not as well articulated. However there is some empirical work.

Given the comprehensive nature of the review I was interested in the qualities of the housing element. Overall I found the housing issue an adjunct to business location. I’m sure the authors would claim that this is likely to be a function of the lack of models to discuss. I notice Dipasquale and Wheaton (1996) did not make the list. The opening line in the abstract uses the word prove. I find that that claim too strong given this paper seeks to highlight the contested terrain. My concern here is the reach of the housing review is not as it could be. Why there was an omission of life cycle models. There may be an inverse relationship between distance and house prices in a standard monocentric urban model but a direct relationship between the two in life cycle models. One would anticipate more expensive sizeable dwellings will be at a distance from the CBD.

On page 6 lines 292 onwards there is an essential concession that for conceptualising issues of location we have to assume a potpuri of issues that are difficult to model eg containing optimisers and satisficers and with discontinuities. I see from the references that calling for an inductive approach is appropriate, which may not fit the current state of housing economics.

Anyway, as it stands, I think this paper is too ambitious and needs to be subdivided. The review of itself is interesting and with more detail could be a useful reference. However, the discussion of finance is necessary in housing which should have a regional dimension. As far as tying that in with EEG etc I suggest that is a logical move but may be a challenge.

The method could also be extended so that the reader is flicking backwards and forwards to understand what is going on. The empirical work though does not have a similar level of detail as the review. I found the empirical work difficult to follow and not very well articulated. Why knn=5? Is it not worrisome that OLS trumps a spatial model when clustering is concluded?

When combining work and home locations in a unified piece, in a sense, what is to be considered is unnecessary commuting, but I think a different literature is needed.

Author Response

Dear Reviewer, please see attached corrected version of the paper with the answers to your remarks. 

best regards

Reviewer 4 Report

The paper is well structured and addresses issues related to the interactions between business and housing location models

The bibliographic references are complete for the subject matter.

The variables used in the six econometric models used are effective in explaining the phenomena investigated.

Some integration are required of the authors:

1] better clarify whether the unit price of residential properties has been related to their state; it is an aspect that can greatly affect results, especially in peripheral housing areas

2] Are the prices collected (36,000) from the database the actual transition prices (real) or are they obtained on the basis of average values?

3] the reliability of the results obtained by the 6 models is to be correlated with the statistical indicators, such as the adjusted RC2; for some models (1, 2 and 3) this indicator seems to be well below the minimum acceptable values. This is an aspect to be reported in the results. It would also be useful to have for each variable, the verification of their significance (t Student) and the general one of the model (F Fisher)

Author Response

Dear reviewer, thank you for your valuable comments. We have addressed all your points in the text and answered them in a table. All changes are visible in the track-mode version, 

best regards, Authors

Round 2

Reviewer 3 Report

Special Issue "Reinvigorating Research on Housing Inequalities and Housing Price Mechanism Using Emerging Data and Technologies"

Manuscript ID land-1435401 Spatial interactions in business and housing location models -2

The second impression of this paper is that it is very similar to the first in the areas where I raised issues.

In line 571 it is asserted that ‘this reflects the strength of spatial patterns readable from data, but as long as the  model is explanatory, not predictive, it is acceptable’. So explanation is the key here not prediction. In line 19 It  also shows that assumptions on unimodal spatial urban structure, the existence of rational maximisers, distance-decaying externalities and a single pattern of behaviour does not hold. This is not tested.

One the maximiser assumption is rejected the next bit is ok..

Instead, one deals in reality with multimodality, a mixture of maximisers and satisfiers, incomplete information, appearance of spatial interactions, feed-back loops. But …the existence of persistence of behaviour with slow and costly adjustments of location.. is a problem. This adjustment to what… an equilibrium? Which implies utility maximisers?

The life cycle points for me concerns Stein (1995) and Ortalo-Magné & Rady (2006). This explains house price volatility. But then it would help explain line 750  ‘From the data analysis, we derived that the number of transactions is the highest in the CBD and also in the cheapest district in the north.’ Why? There will be more transactions in neighbourhoods with smaller dwellings – a family size thing – these will be cheaper and more likely at the periphery. By implication larger houses will be sold less often and be more expensive.  I accept a macro approach to housing where lifecycle issues are ignored but issues keep emerging. I still find finance is such a key issue. I see in line 575 ‘Due to comparability issues, we did not include real estate features’. How does this sit with bonding housing with business profitability?

Dipasquale and Wheaton (1996) now makes the list of refs but is not used well. Defacto, the paper and D&W say the same thing about housing markets being co-defined with labour markets. They avoid quality issues by claiming a stable structure of local house prices  Also income is the key determinant of price. However, income would be a proxy for finance if house price earnings ratios were the same across space, but they are not.

Anyway, as it stands, I think this paper is too ambitious and needs to be subdivided. The review of itself is interesting and with more detail could be a useful reference. However, the discussion of finance is necessary in housing which should have a regional dimension. As far as tying that in with EEG etc I suggest that is a logical move but may be a challenge. Overall I don’t see the coevolutionary ideas in housing in this rewrite.

Ortalo-Magné, F. & Rady, S. (2006) Housing Market Dynamics: On the Contribution of Income Shocks and Credit Constraints, Review of Economic Studies, 73(2), pp. 459-485.

Author Response

This is the second round of review of our paper. In the previous round, we have obtained comments from three reviewers, which was the basis for introducing many changes in the article. One of the reviewers is still not satisfied with improvements and raises a few points. In the team of Authors, we have analysed the comments in detail, and we do not think they should be included in the text as they change the message coming from our study. Please find below the table with our explanations.  
